# A Search for Causes of Rising Incidence of Differentiated Thyroid Cancer in Children and Adolescents after Chernobyl and Fukushima: Comparison of the Clinical Features and Their Relevance for Treatment and Prognosis

**DOI:** 10.3390/ijerph18073444

**Published:** 2021-03-26

**Authors:** Valentina Drozd, Vladimir Saenko, Daniel I. Branovan, Kate Brown, Shunichi Yamashita, Christoph Reiners

**Affiliations:** 1The International Fund “Help for Patients with Radiation-Induced Thyroid Cancer ‘Arnica’”, 220005 Minsk, Belarus; 2Atomic Bomb Disease Institute, Nagasaki University, Sakamoto 1-12-4, Nagasaki 852-8523, Japan; saenko@nagasaki-u.ac.jp; 3New York Ear, Nose and Throat Institute, Project Chernobyl, 1810 Voorhies Avenue, Brooklyn, NY 11235, USA; personal@doctorbranovan.com; 4Program of Science, Technology and Society, Massachusetts Institute of Technology, 77 Massachusetts Avenue, Cambridge, MA 02139, USA; brownkl@mit.edu; 5Global Exchange Center, Fukushima Medical University, Hikarigaoka 1, Fukushima 960-1295, Japan; shun@nagasaki-u.ac.jp; 6Center for Advanced Radiation Medical Center, National Institutes for Quantum and Radiological Science and Technology, 4-9-1 Anagawa, Inageku, Chiba 263-4095, Japan; 7Clinic and Polyclinic of Nuclear Medicine, University of Würzburg, Oberdürrbacher Str. 6, D-97080 Würzburg, Germany; reiners_c@ukw.de

**Keywords:** rising incidence of thyroid cancer, screening and overdiagnosis, pediatric thyroid cancer after Chernobyl and Fukushima, nitrate and thyroid carcinogenesis

## Abstract

The incidence of differentiated thyroid cancer (DTC) is steadily increasing globally. Epidemiologists usually explain this global upsurge as the result of new diagnostic modalities, screening and overdiagnosis as well as results of lifestyle changes including obesity and comorbidity. However, there is evidence that there is a real increase of DTC incidence worldwide in all age groups. Here, we review studies on pediatric DTC after nuclear accidents in Belarus after Chernobyl and Japan after Fukushima as compared to cohorts without radiation exposure of those two countries. According to the Chernobyl data, radiation-induced DTC may be characterized by a lag time of 4–5 years until detection, a higher incidence in boys, in children of youngest age, extrathyroidal extension and distant metastases. Radiation doses to the thyroid were considerably lower by appr. two orders of magnitude in children and adolescents exposed to Fukushima as compared to Chernobyl. In DTC patients detected after Fukushima by population-based screening, most of those characteristics were not reported, which can be taken as proof against the hypothesis, that radiation is the (main) cause of those tumors. However, roughly 80% of the Fukushima cases presented with tumor stages higher than microcarcinomas pT1a and 80% with lymph node metastases pN1. Mortality rates in pediatric DTC patients are generally very low, even at higher tumor stages. However, those cases considered to be clinically relevant should be followed-up carefully after treatment because of the risk of recurrencies which is expected to be not negligible. Considering that thyroid doses from the Fukushima accident were quite small, it makes sense to assess the role of other environmental and lifestyle-related factors in thyroid carcinogenesis. Well-designed studies with assessment of radiation doses from medical procedures and exposure to confounders/modifiers from the environment as e.g., nitrate are required to quantify their combined effect on thyroid cancer risk.

## 1. Introduction: Thyroid Cancer Incidence in Adults

In the last thirty years, the incidence of differentiated thyroid carcinoma (DTC) has been steadily increasing globally and especially rapidly in developed countries such as South Korea, France, Italy, and the United States. For example, in the United States, DTC incidence increased from 4.56 per 100,000 person-years in 1974–1977 to 14.42 per 100,000 person-years in 2010–2013 [1]. For decades, researchers have explained the global rise in DTC as the result of new diagnostic modalities and highly sensitive equipment and screening [2,3]. For this reason, specialists in several countries now recommend against unnecessary screening to avoid overdiagnosis [3]. On the other hand, cancer registries recorded a significant increase not only of small, early stage tumors but larger, later-stage tumors too, which is contrary to the idea of the dominant impact of screening [1,4].

In contrast with the major increases observed for incidence, long-term DTC mortality declined almost everywhere, or stabilized around a value of 0.5/100,000 in both sexes. Only in Canada, the United States, and Australia, the downward trends leveled off or slightly increased since around the 1990s in both sexes [5]. Statistically significant mean annual increases of mortality from 1986 to 2015 were recorded in the United States for both sexes with 0.06% (95% CI: 0.01–0.12) in women and 0.21%, (95% CI: 0.19–0.23) in men, and in Canada in men with 0.08% (95% CI: 0.02–0.13) [5]. Above that, according to a more comprehensive analysis from the Surveillance, Epidemiology, and End Results Registry (SEER), overall DTC incidence-based mortality increased significantly in the USA during 1994–2013 by 1.1% per year (95% CI: 0.6–1.6%), which was mainly related to advanced stage papillary thyroid carcinoma with 2.9% per year (95% CI: 1.1–4.7%) [1]. Recently Yan et al. (2020) performed a retrospective analysis of all 69,684 individuals with DTC reported in the California Cancer Registry (2000–2017) [6]. The authors pointed out that thyroid cancer-specific mortality rates increased on average by 1.7% per year (*p* < 0.001); in men by 2.7% per year (*p* < 0.001) and in cases with larger tumors (2–4 cm) by 3.4% per year (*p* < 0.05). However, the mortality rates of women and patients with tumors ≤1 cm remained stable [6].

With the exception of very small tumors, these troubling indicators suggest that the focus on screening and detection on “early cancers” as the only cause of the upsurge in DTC incidence may obscure other factors that possibly cause aggressive and lethal cancers, which are most likely environmental in nature.

This article reviews relevant studies on pediatric DTC in Belarus (exposed to Chernobyl radiation and without such exposure) and Japan (exposed to extremely low dose radiation from Fukushima and without such exposure). We try to find out possible factors additional to accidental radiation exposure (in particular, medical diagnostic radiation exposure and excessive nitrate intake) that might synergistically contribute to the alarming rise not only of small clinically “indolent” thyroid cancers witnessed in many industrialized countries on the globe. Additionally, comparison of the clinical features and consequences for treatment and prognosis shall provide a better understanding of the role of screening as a potential trigger for increased incidence.

## 2. Trends in Pediatric Thyroid Cancer Incidence Worldwide

Differentiated thyroid carcinoma accounts for 2–4% of all pediatric malignancies and is a relatively rare tumor. The incidence of thyroid cancer in children below 14 years of age is 0.5–1.2/million and 4.4–11/million for adolescents between 15 and 19 years of age, with constantly growing number of cases in both Europe and America [7,8].

Trends in pediatric thyroid cancer incidence in the United States were studied by Bernier et al. (2019) among 7296 cases of 0–19 year-old children and adolescents using data from 39 U.S. cancer registries during 1998–2013. Age-standardized incidence rates (ASR) of pediatric DTCs significantly increased by 4.4%/year (95% CI: 3.74–5.13). Annual percent change (APC) of DTC incidence were highest for small tumors sizes <1 cm = +9.5%/year (95% CI: 6.13–12.90), intermediate for sizes 1–2 cm = +7.0%/year (95% CI: 4.31–9.60) and lowest for sizes >2 cm = +4.7%/year; (95%CI: 2.75–6.67). This observation could be related to a screening effect. However, APC of DTC incidence rates increased significantly over the time period with advancing tumors stages from localized +4.1%/year (95% CI: 2.84–5.29), regional +5.7%/year (95% CI: 4.64–6.73) to distant +8.6%/year (95% CI: 5.03–12.19) [8]. The authors concluded that significantly increasing rates of pediatric DTC are unlikely to be entirely explained by enhanced medical surveillance during childhood as rates of large and late stage DTCs are increasing over time [8].

In a cross-sectional study by Qian et al. (2019), 1806 individuals younger than 20 years with thyroid cancer of the SEER from 1973 to 2013 were analyzed. The main result of this analysis was that incidence of pediatric thyroid cancer gradually increased by 1.1% per year from 1973 to 2006 and markedly increased by 9.6% per year from 2006 to 2013. It should be noted that the incidence of large tumors (>20.0 mm) gradually increased from 1983 to 2006 (APC: 2.23%; 95% CI: 0.93–3.54%) and then significantly increased from 2006 to 2013 (APC: 8.84%; 95% CI: 3.20–14.79%). Additionally, the incidence rates of regionally extended thyroid cancer gradually increased from 1973 to 2006 (APC: 1.44%; 95% CI: 0.68–2.21%) and then significantly increased from 2006 to 2013 (APC: 11.16%; 95% CI: 5.26–17.40%). This significant increase in large and regionally extended tumors indicates that enhanced diagnosis of small, indolent thyroid cancers is not solely responsible for this trend and represents a true increase [9].

Recently, a group of Brazilian researchers [10], based on a study of 11 registries, reported that the age-adjusted incidence rates of thyroid cancer in children (0–14 years) significantly increased from 0.2 in 2000 to 2.8 per million in 2013 with an annual average percent change of 18.8 [95% CI: 8.1–30.6]. The authors concluded that this increased incidence is unlikely to be explained by screening, as children less than 14 years of age do not routinely undergo medical surveillance for thyroid tumors and environmental risk factors must be discussed [10].

This year, Vaccarella et al. (2021) [11] performed a population-based comparison of thyroid cancer incidence in children and adolescents aged 0–19 years (in 49 countries and territories) and mortality (in 27 countries) based on data from the International Incidence of Childhood Cancer Volume 3 study database, the WHO mortality database, and the Cancer Incidence in Five Continents database. Age-standardized incidence rates of thyroid cancer among children and adolescents in 2008–12 ranged from 0.4 (in Uganda and Kenya) to 13.4 (in Belarus) cancers per 1 million person-years. Rapid increases in incidences between 1998–2002 and 2008–2012 were observed in almost all countries and strongly correlated (*r* > 0.8) with rates in adults. Thyroid cancer mortality in children and adolescents was less than 0.1 per 10 million person-years in each country. Contrary to studies cited above, the authors are suggesting a major role for overdiagnosis, which, in turn, can lead to overtreatment [11]. However, in a previous paper on rising thyroid cancer incidence in adults, Vaccarella et al. (2016) gave more concrete estimates about the percentages of cases being presumably overdiagnosed by screening with 70% in Italy, France and South Korea, 40% in the United States and less than 25% in Japan, the Nordic Countries, England & Scotland and Australia [2].

To recapitulate the data from the literature cited above, thyroid cancer incidence in children and adolescents is increasing considerably over time worldwide. On the contrary, thyroid cancer mortality in this age group remains constantly low. However, it seems to be premature to conclude that overdiagnosis by screening plays a major role in this context for several reasons:(1)First, recommendations on prevention of “overdiagnosis” and “overtreatment” of DTC merely on the basis of data from epidemiological registries should not be generalized without taking into account the clinical behavior of this tumor.(2)Experiences with screening in adults show that only round about 50% of thyroid cancers detected with screening are early, “indolent” tumors.(3)Disproportional increases of incidences in larger, regionally extended and later-stage tumors in pediatric DTC demand for adequate diagnostic and therapeutic concepts.(4)Even if mortality of DTC in children and adolescents is low, the risk of recurrencies in those clinically relevant tumors with disproportionate increases should not be trivialized.

## 3. Pediatric Thyroid Cancer after the Chernobyl Reactor Accident in Belarus

It is well known that exposure to radiation to atomic bomb survivors and those exposed by therapeutic procedures is linked to an increased risk for DTC specifically in children. However, when Belarusian scientists first reported in *Nature* a surge in pediatric DTC 4 years after the Chernobyl accident in April 1986 [12], a dozen international experts disputed this finding [13,14]. They attributed the increases to mass screening and local iodine deficiency.

Today, the high increase in the incidence of thyroid cancer among children, who were diagnosed in Belarus between 1986 and 1996 after the Chernobyl accident from 0.5/ up to 30/1 million/ year is well accepted. In the areas of Belarus most heavily contaminated with radionuclides, the incidence even increased to 90 cases/million/year [15,16].

In detail, data obtained from different screening programs by both international and local organizations in contaminated areas of Belarus (1990–2000) showed variations of childhood DTC prevalence between 0.2% and 0.6% in Gomel, 0.3% in Brest and 0.008% in Mogilev [17,18,19,20]. In 1996, an UNSCEAR committee acceded that the “DTC epidemic” was real and that among children under age 18 in 1986 was largely caused by internal exposure to radioactive iodine due to consumption of contaminated milk and food. Between 1991 and 2005, 6848 cases of childhood DTC were reported [21]. The increase of thyroid cancer cases began to appear about 4–5 years after the accident, persisted up until 2005 and was most expressed among the children under age 10 years at the time of the accident [21,22,23].

Therefore, the main health effect of the Chernobyl accident in the population is an increase in the incidence of thyroid cancer in children. The assessment of radiation doses to the thyroid confirms the link with exposure to radioiodine. Individual thyroid doses due to ^131^I depended on the region of exposure, cow’s milk consumption and the age of the person. Population-averaged thyroid doses among children of youngest age reached from some mGy up to 750 mGy in the most contaminated area, the Gomel Oblast of Belarus [24]. Uncertainties of dosimetry varied in range from 1.6 for doses based on individual-radiation measurements to 2.6 mean geometric standard deviations of individual stochastic doses for “modelled” doses [21,24].

Concerning the clinical relevance of DTC in children detected after Chernobyl, it is necessary to consider clinicopathological characteristics of patients exposed and non-exposed to radiation (sporadic cases). Data from 936 cases of radiation-related DTC in Belarus after Chernobyl [25] and 127 non-exposed cases [26] were published as summarized in Table 1. Radiation-related carcinomas as compared to sporadic tumors were diagnosed significantly more often in boys than in girls (36% vs. 19%), in children as compared to adolescents (56% vs. 34%), and displayed significantly more frequently clinicopathological features of higher tumor aggressiveness with extrathyroidal extension 57% vs. 46% and follicular dominant structural component (48% vs. 31%). Table 1 also shows that distant metastases (to the lung) were significantly more frequent in radiation-related cases (11%) as compared to sporadic ones (2%). The mean tumor diameter amounted to 14.4 mm (range 1–124). With respect to the prevalence of microcarcinomas (smaller than 1.0 cm), the percentages of radiation exposed cases (41%) and sporadic cases (38%) did not differ significantly. Lymph node metastases were very frequent with 74% and 72%, respectively, in both groups. In the Chernobyl group, the proportion of stage pN1b cases showing up with metastases in the lateral neck lymph nodes was relatively high with 40.4% as compared to central lymph nodes stage pN1a with 33.4%. In non-exposed patients, this disproportion was even more pronounced (N1a-22.8%; N1b-48.8%, not shown in Table 1).

The post-Chernobyl pediatric patients were operated in 69% with total thyroidectomy and in 31% with subtotal or hemithyroidectomy. Repeated operations due to relapses had to be performed in 21% of patients, and more than 3 operations were necessary in 2.2% [25,27]. Radioiodine therapy (RAI) was carried out in 69% of patients after total thyroidectomy [25,27].

Late results of follow-up of the Chernobyl childhood thyroid carcinomas show that in spite of its clinical behavior, 20-year event-free survival and relapse-free survival rates were 87.8% and 92.3%, respectively, with a median follow-up of 15.4 years. Overall survival was 96.9% [25]. Since 1990 till 2014, 21 patients (1.9%) died among them 2 from advanced disease, 3 from secondary malignancies, and 3 from other internal diseases; 7 patients committed suicide, and 6 died due to accidents/traumas. Belarusian experience and analysis of long-term follow-up data show that for locoregional relapses after non-total thyroidectomy, the principal risk factors were age less than 15 years at presentation and multifocal growth of tumors, and for distant relapses–the lateral neck lymph node metastases [25,27].

During the follow-up, up to 50% of patients have a number of different health problems related to thyroid cancer treatment such as hypoparathyroidism (12–36% with respect the type of surgery), laryngeal nerve palsy (permanent in 6%, transient in 2%), difficulties of long-term LT4 replacement therapy (up to 40% of patients), RAI side effects (salivary gland dysfunction 44.8%, xerostomia 36%), or depressive states (38%) [28,29,30,31,32]. Recently, a case–control study among females developing DTC after the Chernobyl accident in Belarus ≤19 years at the time of thyroid surgery compared reproductive health indicators over a 20-year follow-up period in patients given RAI (*n* = 111) and controls not given RAI (*n* = 90). Among RAI patients, 78% of cases versus 93% control had a history of pregnancy (*p* < 0.01), and the mean number of pregnancies was 1.5 ± 1.2 in RAI patients versus 1.9 ± 1.1 in controls (*p* < 0.05). It should be noted that infertility was diagnosed in 23% of RAI patients and only in 4% of controls (*p* < 0.01) [33].

Long-term overall survival was excellent in Chernobyl children with DTC from Belarus with 97% after 15 years, however recurrencies occurred in 28% [25,30]. It is well known that risks to develop a second primary malignancy (SPM) after RAI are increased for the gastrointestinal tract (salivary glands, stomach, colorectum), the genitourinary tract (kidneys, bladder, uterus), and the hematopoietic system (leukemia) [34,35,36]. The Belarusian experience shows that among patients with radiation-related thyroid cancer who were operated and treated with radioiodine in childhood, the overall incidence of SPM during follow-up of 15 years is roughly 1% (comprising hematological, cervical, breast, and colonic malignancies) [28]. This percentage seems to be surprisingly low, but patients are still relatively young and longer follow-up of those patients is needed.

To summarize the experiences with Chernobyl-related pediatric thyroid cancer, there is no doubt that radiation played a major role in pathogenesis of these tumors, which can be characterized briefly as follows:(1)Incidence increased first after 4-year latency in the youngest age group 0–9 years at the time of the accident, and the increase was highest in this age group by a factor of appr. 50.(2)High radiation doses to the thyroid up to 750 mGy were mainly caused by ingestion of contaminated milk and food.(3)Clinically, radiation-related childhood cancers differed from sporadic ones mainly by a higher incidence in boys and children of youngest age, extrathyroidal tumor extension and distant metastases. On the other hand, lymph node metastases were very frequent both in radiation-exposed and sporadic cases.(4)In spite of signs of aggressive tumor behavior, 15 years overall survival in radiation-related cases after Chernobyl with more than 95% is excellent. However, quality of life of the patients may be restricted considerably by tumor recurrencies in appr. 30% of patients.(5)Roughly 50% of the patients suffer from treatment-related side effects, which more or less are unavoidable if overall high long-term survival is the treatment goal specifically in patients with advanced tumor stages.

## 4. Pediatric Thyroid Cancer after the Fukushima Nuclear Accident in Japan

In October 2011, seven months after the meltdown of three reactors at the Fukushima Daiichi Nuclear Power Plant, Japanese authorities started the Fukushima Health Management Survey, including thyroid ultrasound examinations, targeting some 370,000 people age 18 and younger [37,38]. The baseline incidence of childhood and adolescent DTC by ultrasound screening in Fukushima was surprisingly high with 116 cases per 300,473 examinees (0.038%) for the first-round examination by March 2014 [38]. Later, from 126 patients who underwent surgery at Fukushima Medical University Hospital by September 2016, 119 cases of papillary thyroid cancer were available for review [39,40].

Because the estimated average radiation doses to the thyroid vary by two orders of magnitude between Fukushima (a few mSv) and Chernobyl (several hundred mSv) experts attributed the Fukushima cancers to the population-based screening program [41,42].

Definitely, screening should have played an important role; in support—typically for a screening effect—the detection rate in three screening rounds of the Fukushima Health Management Survey decreased by a factor of ten over the time of 7 years [41]. As compared to Chernobyl, Fukushima childhood cancers cases detected by screening appeared immediately after the accident and not with a latency of 4–5 years which was the case after Chernobyl, which again is considered as an argument against radiation exposure as the main cause of Fukushima thyroid cancer cases detected by screening [42]. However, despite of the screening, it should be considered that most cases of cancer in Fukushima were clinically not “indolent” and would presumably have been diagnosed sometime later because of clinical symptoms (as e.g., lumps in the neck or enlarged lymph nodes), which then need to be treated.

In 2019, Suzuki et al. from Fukushima Medical University published their experiences with surgical treatment of 115 cases of DTC in children adolescents detected by population-based screening after the Fukushima reactor accident [40]. Mean age was 17.8 years (range 9–23); 64% of the cases were female and 36% male. The mean tumor diameter amounted to 14.8 mm (range 6–51); only 17.4% of the cases were staged as microcarcinomas stage pT1a. By the way, the TNM classification does not take into account the specific anatomy of the thyroid in children (TNM 7th edition). The pT1a category of a tumor <1 cm diameter is defined for healthy adults with a normal thyroid volume of appr. 20 mL. In contrast, for a 10-year old child, with a normal thyroid volume of appr. 8 mL, the pT1a limit should be reduced to 0.4 cm! [43].

Of note, 42.1% of the tumors operated at Fukushima University Medical center [40] presented with extrathyroidal spread to fat and connective tissue of the neck. Above that, 80% of the 115 cases presented with lymph node metastases (pN1), among them 63.5% with central (pN1a) and 16.5% lateral (pN1b) metastases to the neck lymph nodes [40]. Suzuki et al. compared subgroups of 78 patients diagnosed within 4 years after the accident and 37 more than years after it. They found no differences of the characteristics and assume, that the tumors have a common etiology. This is considered to be important because after Chernobyl, DTC incidence rose with a latency of 4 years which suggests that radiation is not this “common etiology” in Fukushima [40].

In the Fukushima thyroid cancer cases, hemithyroidectomy was performed in 92%, total thyroidectomy in 8%, lymph node dissection of the central compartment in 85% and of the lateral compartment in 15% of all Fukushima cases [40].

For comparison, data from non-radiation exposed children and adolescents with thyroid cancers from Tokyo Ito hospital can serve [44]. Mean age of these 153 cases (136 females, 17 males) was 16 years (range 7–18). Tumors were larger with a mean diameter of 25 mm (interquartile range IQR 17–40). Primary tumor size ≤10 mm was diagnosed in 7.2% of cases.

The children and adolescents with DTC from Tokyo showed lymph node metastases in 73.2%, among them stage N1a in 20.9% and pN1b in 52.3%, Interestingly, the lymph node positivity did not depend on tumor size. Gross extrathyroidal extension was found in 5.9%. Total thyroidectomy was performed in 24% and neck dissection in 83% of the patients. The median follow-up period of this study was 14.8 years; no patient died of the disease but 22.2% of patients developed recurrences, among them in 5.8% secondaries to the lung [44]. The 10-, 20-, and 30-year disease-free survival rates were 83.8%, 71.7%, and 53.5%, respectively. Multivariate analysis revealed the following risk factors (hazard ratios HR) related to worse disease-free survival: extrathyroidal growth HR = 7.04 (*p* < 0.005), more than 10 metastatic lymph nodes HR = 3.49 (*p* < 0.005) and clinically detectable lymph node metastases cN1 HR = 3.40 (*p* < 0.005). In addition, applying univariate statistics, lateral lymph node metastases pN1b were associated with poorer survival (HR = 4.35, *p* < 0.005) [44].

To summarize the experiences with Fukushima-related pediatric thyroid cancer cases, there is no proof that radiation played a major role in pathogenesis of these tumors:(1)Low radiation doses to the thyroid between 10 mGy and 50 mGy as compared to Chernobyl were due to considerably lower emissions from the damaged reactors affecting the population as well as evacuation and ban of contaminated food and milk.(2)The incidence was relatively high immediately after the beginning of the population-based screening without a latency time, different from Chernobyl.(3)Fukushima related pediatric thyroid cancer cases were older as compared to the Chernobyl patients.(4)Clinically, the percentage of Fukushima microcarcinomas pT1a (17.4%) seemed to be lower as compared to Chernobyl cases (41%), but higher as compared to cases from Tokyo (7.2%).(5)It is too early for any reliable conclusions on the prognosis of pediatric thyroid cancer cases detected by population-based screening in Fukushima. It has to be investigated, if prognostic indicators, which are derivable from Japanese children and adolescents operated in Tokyo, are applicable for Fukushima patients too.(6)Anyhow, it should be very clear that a large proportion of thyroid cancers detected by Fukushima population-based screening cannot be dismissed as “indolent” not needing appropriate treatment and follow-up.

## 5. Other Factors Potentially Contributing to the Increasing Incidence of Thyroid Cancer

Observations from Belarus, Japan and all over the world lead to the question if there are other factors in addition to a screening effect behind the high prevalence of clinically relevant DTC in the population. Maybe those even low doses of radiation by diagnostic imaging at early ages or exposure to environmental hazards might trigger thyroid cancers in children.

A very large study by Hong et al. (2019) of 12 million youths in Korea demonstrated an increased thyroid cancer risk (2.19; 95% CI: 1.97–2.20), including DTC in children, with histories of low doses of radiation from diagnostic procedures [45]. In the U.S between 1996 and 2005, the use of CT scans tripled for children 5 to 14 years of age and this might have contributed to rising pediatric DTC rates [46]. However, especially in young patients, it is good clinical practice not to expose somebody to radiation for screening or diagnosis of trivial diseases. Therefore, when epidemiological studies describe an increased thyroid cancer risks related to diagnostic radiation exposure, the “reverse causation” [47] has to be excluded in the sense that (1) DTC detected was not the indication for radiological imaging and, more importantly, (2) the indication for imaging was not another cancer disease (e.g., tumor of the brain), which may predispose the patient for thyroid cancer if this tumor had been treated previously with radiotherapy. In support in the British Childhood Cancer Survivor Study, the standardized incidence ratio for thyroid cancer as second primary malignancy was highly significantly elevated 18.0 (95% CI: 13.4–23.8) in patients after radiotherapy of brain tumors [48].

More evidence from radiation and molecular biology is needed to better combine epidemiology of DTC with biology-based models of carcinogenesis [49].

Recently published meta-analyses to estimate the association between thyroid cancer risk and iodine intake demonstrated considerable controversy [50,51,52]. Cao et al. (2017) found that excess iodine intake with water in China (>300 mg/d) correlated with reduced risk of thyroid cancer (OR = 0.74, 95% CI, 0.60–0.92) and so was considered as a protective factor [51]. However, another meta-analysis by Lee et al. (2017) found a positive association between thyroid cancer risk and iodine exposure (OR = 1.41 (95% CI, 1.05–1.90) [52]. Presumably these conflicting observations are related to more sensitive diagnosis and living conditions and not to iodine supply [53].

Comorbidity and individual susceptibility may also be considered as etiologic factors of thyroid cancer. A meta-analysis of twelve prospective observational studies by Liang et al. (2018) showed an association of benign thyroid diseases (goiter, hyperthyroidism, thyroiditis) with thyroid cancer risk [54] that cannot be completely associated with greater medical surveillance [55]. A relatively strong association for genetic susceptibility to thyroid cancer was disclosed for rs965513 (*PTCSC2/FOXE1*) on chromosome 9q22.33 [56]. Obesity is a well-described risk factor for thyroid cancer [57,58]. Schmid et al. (2015) published a meta-analysis and demonstrated that the thyroid cancer risk in overweight persons is 25% greater and in individuals with obesity with 55%, respectively, than in their normal weight peers [58]. Several studies have reported, surprisingly, a protective effect of smoking, which may be associated with estrogen metabolism [59,60].

In recent decades, more and more information has been published on the negative health impact of “endocrine disruptors”, environmental pollutants like pesticides, asbestos, benzene, polychlorinated biphenyls (PCB), formaldehyde, polyhalogenated aromatic hydrocarbons (PHAHs), bisphenol A (BPA), and nitrates.

Nitrate has to be considered for as a major and ubiquitous environmental factor, which is used worldwide in exponentially increasing amounts during the past 60 years as crop fertilizer and for food preservation. Nitrates can accumulate for years in both ground and surface water and also pollute the air until the concentration is harmful to human health [61,62,63,64,65]. A meta-analysis of Xie et al. (2016) examined data from 62 observational studies, 49 studies for nitrates and 51 studies for nitrites, including a total of 60,627 cancer cases. The authors showed that dietary nitrite intake was positively associated with thyroid cancer risk with pooled RR of 1.52 (95% CI: 1.12–2.05) [66].

Ecological studies in Belarus after the Chernobyl accident suggest that drinking water with high nitrate content could potentiate the effect of radiation dose and might have affected the rates of DTC in exposed children. According to the official statistics in Belarus from 1960 to 1990, the average nitrate level in drinking water rose sharply about 40-fold, from 1.1 to 41.6 mg/L. The reason for this increase was the intensification of the use of nitrate fertilizers in agriculture from 4 to 92 kg/hectare [20]. Among 1044 cases of pediatric radiation-related DTC from Belarus, the thyroid cancer incidence was significantly correlated with the radiation dose (*p* = 0.029), but the effect of radiation was influenced significantly by the level of nitrates in local drinking water (*p* = 0.004) [20].

The combined influence of Chernobyl related radiation and nitrates particularly in rural residents is assumed to increase the risk for thyroid cancer [67]. Proper epidemiological research like well-designed case–control studies are essentially needed in the future to explore this interrelation.

In contrast to Belarus and Ukraine affected by the Chernobyl accident, the nitrate concentration in water and food is strictly regulated in Japan and did not exceed the reference range at the time of the Fukushima accident [68]. Orita et al. (2015) reported the result of nitrate concentrations in drinking water in Fukushima prefecture. In Kawauchi village, the average nitrate-N level was in the normal range 0.62 mg/L (0.20–2.51 mg/L). The authors concluded that proper studies of dietary nitrate intake are also required to better understand the high incidence of pediatric DTC detected by Fukushima population-based screening [68]. In such studies, not yet established factors (such as excess dietary iodine supply, overweight, hormonal effects, inflammatory conditions, endocrine disruptors and environmental factors) could have been involved.

(1)To summarize the role of potential carcinogenic factors in addition to high doses of radiation—as involved after the Chernobyl accident—a number of confounders may play a role.(2)It seems not to be probable that low doses of radiation *per se* applied for medical diagnostics can induce thyroid cancer in childhood. However, this may be the case if such radiation is combined with confounders.(3)Among those, attention should be spent to nitrate, which is known to increase the risk as such for different cancers (among them thyroid cancer moderately).(4)New data from Belarus lead to the suspicion that nitrate in drinking water increases in combination with radiation the risk for thyroid cancer.(5)Systematic studies are needed to properly investigate this assumption. Such investigations have to take into account other known or suspected confounders in thyroid carcinogenesis—genetic factors, heredity, ethnicity, iodine supply, other thyroid diseases as comorbidity, overweight, smoking, hormones and other endocrine disruptors.

## 6. Conclusions

In conclusion, many authors today argue that the surge in thyroid cancer incidence worldwide—not only in children—is caused by screening or technological advancements in diagnostics, claiming that those cancers are “indolent” leading to “overdiagnosis” and “overtreatment”. We strongly recommend to be cautious and not throw out the baby with the bathwater. As shown in children and adolescents from Belarus and Japan—screened because of radiation exposure or detected as sporadic cases—a considerable proportion of thyroid cancers present as clinically relevant cases larger than microcarcinomas, with extrathyroidal growth and/or with (multiple) lymph node metastases not only in the central compartment. These characteristics have proven to influence disease-free survival of pediatric DTC negatively and therefore have to be taken seriously.

Systematic studies should be carried out to better understand the role of multiple carcinogenic factors as possible drivers of the DTC epidemic, accompanied by use of individualized strategies for risk adapted treatment of childhood thyroid cancer.

## Figures and Tables

**Table 1 ijerph-18-03444-t001:** Clinicopathological characteristics of thyroid cancer in children and adolescents from Belarus (post-Chernobyl and sporadic cases).

	BY Chernobyl	BY Sporadic	*p*-Value
	*n* = 936	*n* = 127	
Reference	[25]	[26]	
Sex F/M (%M)	600/336 (36)	103/24 (19)	<0.001
Children/Adolescents (%Children)	521/415 (56)	43/84 (34)	<0.001
Microcarcinoma (%)	386 (41)	48 (38)	0.501
Lymph node metastasis, N1 (%)	691 (74)	91 (72)	0.593
Distant metastasis, M1 (%)	104 (11)	3 (2)	<0.001
Multifocality (%)	60 (6)	6 (5)	0.560
Extrathyroidal extension, any (%)	387/674 (57)	46/101 (46)	0.031
Dominant structural component			
Papillary (%)	300 (32)	64 (50)	<0.001
Follicular (%)	453 (48)	40 (31)	<0.001
Solid (%)	183 (20)	23 (18)	0.811

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
