# Peer review of "A Search for Causes of Rising Incidence of Differentiated Thyroid Cancer in Children and Adolescents after Chernobyl and Fukushima: Comparison of the Clinical Features and Their Relevance for Treatment and Prognosis"

_ijerph, 2021, doi:10.3390/ijerph18073444_

Round 1

Reviewer 1 Report

This review discusses the causes of upsurge of differentiated thyroid cancer in children and adolescent, and focuses on clinical evidence in Belarus and Fukushima.  Detailed description is very useful how etiology argument of thyroid cancer have evolved since Chernobyl accident occurred. Some description should be  clarified to avoid misleading understanding as indicated below.   Specific comments: 1 Etiology of radiation exposure In Chernobyl, excess thyroid cancer incidence is attributed to radioiodine ingestion according to UNSCEAR. Dose-response relationship has been revealed in thyroid of children. These key information is needed to clarify the etiology together with the similar excess relative risk of atomic-bomb survivors. However, the authors said some questions remain. Excess incidence per thyroid dose varied by regions. It should be recognized that thyroid dose assessment has uncertainty, see https://pubmed.ncbi.nlm.nih.gov/26207684/ It may be better to note that statistical discussion of doses will be needed to clarify your questions.    2 Etiology of medical exposure It is very cautious to refer Hong' s paper that investigated association with medical exposure. Medical exposure to CT in children would be biased by confounding factors or possible reverse causation. The medical reasons should be considered.  See https://pubmed.ncbi.nlm.nih.gov/32556632/.  Head CT in children is unlikely to contribute to excess exposure of thyroid.  It may be better to address these questions.   3 Etiology of Nitrate It may be the first discussion that additional environmental factors are likely to cause excess thyroid cancer incidence.  It would be better to organize several discussion for clarity.   4 Conclusins The following conclusion appears to go too much based on the present review.  ' the same mistake' is not clear.  If your message will be addressed that further follow-up studies of Chernobyl consequence are needed, these should be more clearly described.   ' In conclusion, rather than asserting that the surge in thyroid cancers is caused by a screening effect or technological advancements, and claiming that those cancers are, anyway, “curable”, we should not make the same mistake twice like after Chernobyl. It is very important to take into account the experience of mitigation of the Chernobyl consequences and follow up of patients with radiation-related thyroid cancer in Belarus.'   5 References Ref.34. Full description, volume, page, year  

Author Response

Answers for reviewer 1

  1. This review discusses the causes of upsurge of differentiated thyroid cancer in children and adolescent, and focuses on clinical evidence in Belarus and Fukushima.  Detailed description is very useful how etiology argument of thyroid cancer have evolved since Chernobyl accident occurred. Some description should be  clarified to avoid misleading understanding as indicated below.  

Specific comments: 1 Etiology of radiation exposure In Chernobyl, excess thyroid cancer incidence is attributed to radioiodine ingestion according to UNSCEAR. Dose-response relationship has been revealed in thyroid of children. These key information is needed to clarify the etiology together with the similar excess relative risk of atomic-bomb survivors. However, the authors said some questions remain. Excess incidence per thyroid dose varied by regions. It should be recognized that thyroid dose assessment has uncertainty, see https://pubmed.ncbi.nlm.nih.gov/26207684/ It may be better to note that statistical discussion of doses will be needed to clarify your questions.   

            Our answer: We changed the title of the article to focus more on the clinical features: “A Search for Causes of Rising Incidence of Differentiated Thyroid Cancer in Children and Adolescents after Chernobyl and Fukushima: comparison of the clinical features and their relevance for treatment and prognosis” .

            We changed the structure of the article, highlighted a section with an analysis of the growth in the incidence of thyroid cancer in childhood in the world. We have added a summary for each section. We added more information related to thyroid dose from UNSCEAR 2008 Report and the article by Drozdovich V. “Radiation Exposure to the Thyroid After the Chernobyl Accident. Front Endocrinol (Lausanne). 2021 Jan 5;11:569041. doi: 10.3389/fendo.2020.569041. PMID: 33469445; PMCID: PMC7813882”.

  1. Etiology of medical exposure It is very cautious to refer Hong' s paper that investigated association with medical exposure. Medical exposure to CT in children would be biased by confounding factors or possible reverse causation. The medical reasons should be considered.  Seehttps://pubmed.ncbi.nlm.nih.gov/32556632/.  Head CT in children is unlikely to contribute to excess exposure of thyroid.  It may be better to address these questions.

            Our answer: We have focused on recent publications on the risks of medical exposure, have added information of some important publications, and made conclusions more carefully: “It seems not to be probable that low doses of radiation per se applied for medical diagnostics can induce thyroid cancer in childhood. However, this may be the case if such radiation is combined with confounders”.

  1. Miglioretti, D.L.; Johnson, E.; Williams, A.; Greenlee, R.T.; Weinmann, S.; Solberg, L.I.; Feigelson, H.S.; Roblin, D.; Flynn, M.J.; Vanneman, N. & Smith-Bindman, R. The use of computed tomography in pediatrics and the associated radiation exposure and estimated cancer risk. JAMA pediatrics. 2013, 167(8), 700–707. https://doi.org/10.1001/jamapediatrics.2013.311
  2. Koterov, A.N.; Ushenkova, L.N. & Biryukov, A.P. Hill’s Temporality Criterion: Reverse Causation and Its Radiation Aspect. Biol Bull Russ Acad Sci 47, 1577–1609 (2020). https://doi.org/10.1134/S1062359020120031Han, M.A.; Kim, J.H. Diagnostic x-ray exposure and thyroid cancer risk: systematic review and meta-analysis. Thyroid. 2018. Feb https://doi.org/10.1089/thy.2017.0159.
  3. Taylor, A. J.; Croft, A. P.; Palace, A. M.; Winter, D. L.; Reulen, R. C.; Stiller, C. A.; Stevens, M. C. & Hawkins, M. M.. Risk of thyroid cancer in survivors of childhood cancer: results from the British Childhood Cancer Survivor Study. International journal of cancer. 2009.125(10), 2400–2405. https://doi.org/10.1002/ijc.24581

  1. Etiology of Nitrate It may be the first discussion that additional environmental factors are likely to cause excess thyroid cancer incidence.  It would be better to organize several discussion for clarity.

            Our answer: we have prepared a section 4 where, along with nitrates, we discuss the influence of other factors: “Other factors potentially contributing to the increasing incidence of thyroid cancer”

  1. Conclusions The following conclusion appears to go too much based on the present review.  ' the same mistake' is not clear.  If your message will be addressed that further follow-up studies of Chernobyl consequence are needed, these should be more clearly described.   ' In conclusion, rather than asserting that the surge in thyroid cancers is caused by a screening effect or technological advancements, and claiming that those cancers are, anyway, “curable”, we should not make the same mistake twice like after Chernobyl. It is very important to take into account the experience of mitigation of the Chernobyl consequences and follow up of patients with radiation-related thyroid cancer in Belarus.'  

Our answer: We have prepared a new text for Conclusion.

In conclusion, many authors today argue that the surge in thyroid cancer incidence worldwide – not only in children - is caused by screening or technological advancements in diagnostics, claiming that those cancers are “indolent” leading to “overdiagnosis” and “overtreatment”. We strongly recommend to be cautious and not through out the baby with the bath tub. As shown in children and adolescents from Belarus and Japan – screened because of radiation exposure or detected as sporadic cases – a considerable proportion of thyroid cancers present as clinically relevant cases larger than microcarcinomas, with extrathyroidal growth and/or with (multiple) lymph node metastases not only in the central compartment. These characteristics have proven to influence disease-free survival of pediatric DTC negatively and therefore have to be taken seriously.

Systematic studies should be carried out to better understand the role of multiple carcinogenic factors as possible drivers of the DTC epidemic, accompanied by use of individualized strategies for risk adapted treatment of childhood thyroid cancer.

  1. References Ref.34. Full description, volume, page, year

Our answer: We did it.  Drozd V, Schneider R, Platonova T, Panasiuk G, Leonova T, Oculevich N, Shimanskaja I, Vershenya I, Dedovich T, Mitjukova T, Grelle I, Biko J, Reiners C. Feasibility Study Shows Multicenter, Observational Case-Control Study Is Practicable to Determine Risk of Secondary Breast Cancer in Females With Differentiated Thyroid Carcinoma Given Radioiodine Therapy in Their Childhood or Adolescence; Findings Also Suggest Possible Fertility Impairment in Such Patients. Front Endocrinol (Lausanne). 2020 Oct 28;11:567385. doi: 10.3389/fendo.2020.567385. PMID: 33193085; PMCID: PMC7655975.

Reviewer 2 Report

According to the title and as described in the last paragraph of the Introduction, the manuscript is a review of relevant literature on additional etiological causes which could modify radiation-related risks of differentiated thyroid cancer  (DTC) in children and adolescents after Chernobyl and Hiroshima, and could be responsible for rising thyroid cancer incidence worldwide. In fact, the manuscript focused on a comparison of clinical and pathological features of radiation-related and sporadic thyroid cancers (TC) in paediatric patients in Belarus, as well as in TC patients in Fukushima and Tokyo. Much attention is given to the methods of treatment, its outcomes and complications, and long-term TC patients' survival.  The authors reviewed in detail TC clinicopathological characteristics associated with survival prognosis which is an important questions per se but not related to the review's goal. As for potential TC risk factors other than accidental exposure to ionising radiation, the authors mentioned medical diagnostic procedures and nitrates in drinking water.  Other factors that could be important and relevant for non-exposed populations to explain globally rising TC incidence, just briefly summarized in in pre-last paragraph on page 7. it is strongly recommended to revise the paper focusing on etiological factors that could contribute to TC increase worldwide, or, that is also an important piece of information, summarize up-to-date findings on radiogenic TC in comparison to sporadic ones. The paper should also be revised to keep the information that has relevance and importance for TC risk. To reviewer's opinion, the discussion on p.3 regarding established Chernobyl health consequences and necessity of a life-time follow-up similar to Life Span Study in Japan has no direct relevance to the aim of the review, and could be substantially shortened. The authors didn't mention in their review Belarusian and Ukrainian thyroid cancer and other thyroid diseases screening cohorts. Also they didn't discuss the levels of thyroid exposure in children and adolescence after Chernobyl accident. There is no review of studies on iodine deficiency and TC risks. The authors presented in very detailed manner the results of their own studies. It would be useful to include in the review other teams' recent publications on TC and obesity, genetic predisposition factors, iodine deficiency etc. if the aim of review is to identify factors responsible for rapidly increasing TC rates.

Specific comments:

Table 1:  Unclear what p<0.001 represents in the row "Dominant structural component". please delete or explain

p.5: please spell out "TUE" abbreviation

p.5: why were only 59 TCs included in the comparison, when in the previous para 115 cases were mentioned?        

p.6: reference to the linear non-threshold model when the risk analysis with risk expressed in odd ratios (OR) is based on the fact/ type of medical diagnostic procedure doesn't look relevant in this case. Suggest to revise/ remove the sentences starting from "...that these risk estimates were obtained with the idea of a linear non-threshold model...." till the end of the paragraph.

p.6: A paragraph on nitrate metabolism could be substantially shortened keeping the information relevant to TC. Citation of glioma risk estimate is not relevant to the review. Please, revise the paragraph.   

Author Response

Answers for reviewer 2

  1. According to the title and as described in the last paragraph of the Introduction, the manuscript is a review of relevant literature on additional etiological causes which could modify radiation-related risks of differentiated thyroid cancer  (DTC) in children and adolescents after Chernobyl and Hiroshima, and could be responsible for rising thyroid cancer incidence worldwide. In fact, the manuscript focused on a comparison of clinical and pathological features of radiation-related and sporadic thyroid cancers (TC) in paediatric patients in Belarus, as well as in TC patients in Fukushima and Tokyo. Much attention is given to the methods of treatment, its outcomes and complications, and long-term TC patients' survival.  The authors reviewed in detail TC clinicopathological characteristics associated with survival prognosis which is an important questions per se but not related to the review's goal.

            Our answer: We changed the title of the article to focus more on the clinical features: “A Search for Causes of Rising Incidence of Differentiated Thyroid Cancer in Children and Adolescents after Chernobyl and Fukushima: comparison of the clinical features and their relevance for treatment and prognosis”. We changed the structure of the article, highlighted a section with an analysis of the growth in the incidence of thyroid cancer in childhood in the world. We have added a summary for each section. We changed the second section and added publications of recent years on the incidence of thyroid cancer in children “Trends in pediatric thyroid cancer incidence”

  1. Bernier, M.O.; Withrow, D.R.; Berrington de Gonzalez A.; Lam C.; Linet, M.S.; Kitahara, C.M. & Shiels, M.S. Trends in pediatric thyroid cancer incidence in the United States, 1998-2013. Cancer. 2019. 125(14), 2497–2505. https://doi.org/10.1002/cncr.32125
  2. Qian, Z.J.; Jin, M.C.; Meister, K.D. & Megwalu, U.C. Pediatric Thyroid Cancer Incidence and Mortality Trends in the United States, 1973-2013. JAMA otolaryngology-- head & neck surgery. 2019, 145(7), 617–623. https://doi.org/10.1001/jamaoto.2019.0898
  3. de Souza Reis, R.; Gatta, G. & de Camargo, B. Thyroid carcinoma in children, adolescents, and young adults in Brazil: A report from 11 population-based cancer registries. PloS one. 2020, 15(5), e0232416. https://doi.org/10.1371/journal.pone.0232416
  4. Vaccarella, S.; Lortet-Tieulent, J.; Colombet, M.; Davies, L; Stiller, C. A.; Schüz, J.; Togawa, K.; Bray, F.; Franceschi, S.; Dal Maso, L.; Steliarova-Foucher, E. & IICC-3 contributors. Global patterns and trends in incidence and mortality of thyroid cancer in children and adolescents: a population-based study. The lancet. Diabetes & endocrinology. 2021, 9(3), 144–152. https://doi.org/10.1016/S2213-8587(20)30401
  5. As for potential TC risk factors other than accidental exposure to ionising radiation, the authors mentioned medical diagnostic procedures and nitrates in drinking water.  Other factors that could be important and relevant for non-exposed populations to explain globally rising TC incidence, just briefly summarized in in pre-last paragraph on page.

            Our answer: We have prepared a section 4 “Other factors potentially contributing to the increasing incidence of thyroid cancer”, where discuss, along with nitrates and medical diagnostic procedures, the influence of other factors. We have added information regarding other risk factors: medical diagnostic procedures, comorbidity and individual susceptibility, iodine intake, obesity, smoking, “endocrine disruptors”, environmental pollutants.

  1. It is strongly recommended to revise the paper focusing on etiological factors that could contribute to TC increase worldwide, or, that is also an important piece of information, summarize up-to-date findings on radiogenic TC in comparison to sporadic ones. The paper should also be revised to keep the information that has relevance and importance for TC risk.

Our answer: We have added more clinical information to compare radiogenic TC to sporadic ones. We made conclusions about the differences between sporadic and radiogenic thyroid cancer in children.

  1. To reviewer's opinion, the discussion on p.3 regarding established Chernobyl health consequences and necessity of a life-time follow-up similar to Life Span Study in Japan has no direct relevance to the aim of the review, and could be substantially shortened.

Our answer: We have significantly reduced and shortened this part of the article.

  1. The authors didn't mention in their review Belarusian and Ukrainian thyroid cancer and other thyroid diseases screening cohorts. Also they didn't discuss the levels of thyroid exposure in children and adolescence after Chernobyl accident.

Our answer: We focused in our article on Belarusian data of patients with thyroid cancer. We added more information related thyroid exposure in children and adolescence after Chernobyl accident: UNSCEAR 2008 Report, UNSCEAR 2020 Report, the last article by Drozdovich V.

  1. United Nations Scientific Committee on the Effects of Atomic Radiation 2011; Sources and Effects of Ionizing Radiation, UNSCEAR 2008 Report to the General Assembly with Scientific Annexes. Annex D: Health effects due to radiation from the Chernobyl accident. New York: United Nations. 219 p.
  2. United Nations Scientific Committee on the Effects of Atomic Radiation 2020; Sources and Effects of Ionizing Radiation, UNSCEAR 2020 Report to the General Assembly with Scientific Annexes. Annex B: Levels and effects of radiation exposure due to the accident at the Fukushima Daiichi Nuclear Power Station: implications of. New York: United Nations. 248 p
  3. Radiation Exposure to the Thyroid After the Chernobyl Accident. Front Endocrinol (Lausanne). 2021 Jan 5;11:569041. doi: 10.3389/fendo.2020.569041. PMID: 33469445; PMCID: PMC7813882
  4. There is no review of studies on iodine deficiency and TC risks. The authors presented in very detailed manner the results of their own studies. It would be useful to include in the review other teams' recent publications on TC and obesity, genetic predisposition factors, iodine deficiency etc. if the aim of review is to identify factors responsible for rapidly increasing TC rates.

Our answer:    We have prepared a 4 section “Other factors potentially contributing to the increasing incidence of thyroid cancer” where discuss, along with nitrates and medical diagnostic procedures, the influence of other factors. We have added information regarding other risk factors: comorbidity and individual susceptibility, iodine intake, obesity, smoking, “endocrine disruptors”, environmental pollutants.

  1. Zimmermann MB,; Galetti V. Iodine intake as a risk factor for thyroid cancer: a comprehensive review of animal and human studies. Thyroid research. 2015 Jun 18; 8:8. doi: 10.1186/s13044-015-0020-8. eCollection 2015.
  2. Cao LZ,; Peng XD,; Xie JP,; Yang FH,; Wen HL, ;Li S. The relationship between iodine intake and the risk of thyroid cancer: A meta-analysis. Medicine (Baltimore). 2017 May; 96(20):e6734. doi: 10.1097/MD.0000000000006734.
  3. Lee JH,; Hwang Y,; Song RY,; Yi JW,; Yu HW,; Kim SJ,; Chai YJ,; Choi JY,; Lee KE,; Park SK. Relationship between iodine levels and papillary thyroid carcinoma: A systematic review and meta-analysis. Head Neck. 2017; 39(8): 1711-1718. doi: 10.1002/hed.24797. Epub 2017 May 17. PMID: 28513893.
  4. Lv, C.; Yang, Y.; Jiang, L.; Gao, L.; Rong, S.; Darko, G. M.; Jiang, W.; Gao, Y. & Sun, D. Association between chronic exposure to different water iodine and thyroid cancer: A retrospective study from 1995 to 2014. The Science of the total environment. 2017. 609, 735–741. https://doi.org/10.1016/j.scitotenv.2017.07.101
  5. Liang L; Zheng XC,; Hu MJ,; Zhang Q,; Wang SY,; Huang F. Association of benign thyroid diseases with thyroid cancer risk: a meta-analysis of prospective observational studies. J Endocrinol Invest.2019 Jun;42(6):673-685. doi: 10.1007/s40618-018-0968-z. Epub 2018 Nov 1.
  6. Kitahara CM,; Farkas DK,; Jørgensen JOL,; Cronin-Fenton D,; Sørensen HT. Benign thyroid diseases and risk of thyroid cancer: a nationwide cohort study. J Clin Endocrinol Metab. 2018; 103(6): 2216–2224.
  7. Gudmundsson, J.; Thorleifsson, G.; Sigurdsson, J. K.; Stefansdottir, L.; Jonasson, J. G.; Gudjonsson, S. A.; Gudbjartsson, D. F.; Masson, G.; Johannsdottir, H.; Halldorsson, G. H.; Stacey, S. N.; Helgason, H.; Sulem, P.; Senter, L.; He, H.; Liyanarachchi, S.; Ringel, M. D.; Aguillo, E.; Panadero, A.; Prats, E., … Stefansson, K. A genome-wide association study yields five novel thyroid cancer risk loci. Nature communications. 2017. 8, 14517. https://doi.org/10.1038/ncomms14517
  8. Steele, C. B.; Thomas, C. C.; Henley, S. J.; Massetti, G. M.; Galuska, D. A.; Agurs-Collins, T.; Puckett, M. & Richardson, L. C. Vital Signs: Trends in Incidence of Cancers Associated with Overweight and Obesity - United States, 2005-2014. MMWR. Morbidity and mortality weekly report. 2017. 66(39), 1052–1058. https://doi.org/10.15585/mmwr.mm6639e1
  9. Schmid D,; Ricci C,; Behrens G,; Leitzmann MF. Adiposity and risk of thyroid cancer: a systematic review and meta-analysis. Obes Rev. 2015; 16(12):1042–1054. 
  10. Kitahara CM,; Linet MS,; Beane Freeman LE,; Check DP,; Church TR,; Park Y,; Purdue MP,; Schairer C,; Berrington de González A. Cigarette smoking, alcohol intake, and thyroid cancer risk: a pooled analysis of five prospective studies in the United States. Cancer Causes Control. 2012 Oct;23(10):1615-24. doi: 10.1007/s10552-012-0039-2. Epub 2012 Jul 29. PMID: 22843022; PMCID: PMC3511822.
  11. Galanti MR.; Hansson L.; Lund E.; Bergström R.; Grimelius L.; Stalsberg H.; Carlsen E.; Baron JA.; Persson I.; Ekbom A. Reproductive history and cigarette smoking as risk factors for thyroid cancer in women: a population-based case-control study. Cancer Epidemiol Biomarkers Prev. 1996 Jun;5(6):425-31. PMID: 8781737

Specific comments:

Table 1:  Unclear what p<0.001 represents in the row "Dominant structural component". please delete or explain

- We deleted

p.5: please spell out "TUE" abbreviation

- the Thyroid Ultrasound Examination

p.5: why were only 59 TCs included in the comparison, when in the previous para 115 cases were mentioned?

- We removed a portion of the table relating to the Japanese data.

p.6: reference to the linear non-threshold model when the risk analysis with risk expressed in odd ratios (OR) is based on the fact/ type of medical diagnostic procedure doesn't look relevant in this case. Suggest to revise/ remove the sentences starting from "...that these risk estimates were obtained with the idea of a linear non-threshold model...." till the end of the paragraph.

- We deleted

p.6: A paragraph on nitrate metabolism could be substantially shortened keeping the information relevant to TC. Citation of glioma risk estimate is not relevant to the review. Please, revise the paragraph.

We deleted “glioma risk estimate”. 

Round 2

Reviewer 2 Report

The manuscript substantially improved after extensive revisions made by the authors. 

This manuscript is a resubmission of an earlier submission. The following is a list of the peer review reports and author responses from that submission.